# OpenReview forum: "THE PERSONALIZATION TRAP: HOW USER MEMORY ALTERS EMOTIONAL REASONING IN LLMS"
_ICLR.cc/2026/Workshop/AFAA — AFAA 2026 Poster_

### Official Review · Reviewer_NFAy · 2026-02-14
**Good Paper With Missing Robustness Checks**

**Rating:** 3
**Confidence:** 3

**Summary:**

This paper investigates how injecting user memory (persistent persona information) into LLM system prompts affects emotional reasoning, evaluated across 15 models using two validated psychometric instruments (STEU and STEM). The authors construct pairs of advantaged and disadvantaged user profiles grounded in Bourdieu's social capital framework, as well as intersectional personas varying across gender, age, religion, and ethnicity. They find that adding user profiles systematically degrades emotional understanding accuracy, with disadvantaged personas causing larger performance drops than advantaged ones, and that demographic attributes produce significant biases in both emotion recognition and emotional guidance tasks. They also explore a DPO-based mitigation strategy on smaller models, showing preliminary improvements without degrading general capabilities.

**Strengths:**

This paper addresses a timely and understudied question. Memory is now integrated into major chatbot products, yet how persistent user context shapes model behavior over time remains poorly understood. This work is a meaningful step toward filling that gap.

The use of validated psychometric instruments (STEU and STEM) rather than ad-hoc benchmarks is a strong methodological choice. These tests have established ground-truth answers grounded in expert consensus, which makes the accuracy comparisons interpretable and the findings harder to dismiss as artifacts of subjective evaluation.

The human annotation pipeline is notably rigorous. Rather than running evaluations on the full STEU and STEM datasets as-is, the authors implemented a two-phase filtering process with a quality gate and persona-sensitivity check, removing any items where persona context could legitimately change the correct answer. The speeding filter and requirement of nine valid annotations per item further strengthen confidence in the filtered dataset. I hope the authors share this curated dataset so others can build on it.

The persona construction is also carefully done. The 93% annotator preference for enriched personas over real self-reported PRISM profiles gives confidence that the experimental stimuli are realistic, and the controls on length difference and negative sentiment across advantaged/disadvantaged pairs help rule out obvious confounds.

The breadth of the evaluation across 15 models, spanning multiple architectures, scales, and reasoning modes, adds substantial generalizability. The finding that reasoning-capable models show lower biases than their standard counterparts is a useful practical insight.

Finally, the authors test multiple memory injection methods (system prompt, RAG, long-context, conversation history), which partially addresses concerns about results being artifacts of a single prompting strategy, though a major gap remains as discussed below.

**Weaknesses:**

The paper's central claim is that persona-specific information drives the observed performance shifts. However, Section 5.2 demonstrates that simply prepending unrelated conversational or long-context material also degrades STEU accuracy substantially (e.g., Claude 3.5 Haiku drops from 69.05% to 43.57% with early conversation injection). This raises a serious confound: would injecting persona-irrelevant system prompt content of comparable length, such as "It is sunny outside," "Today is Sunday," or random factual statements, produce a similar magnitude of flips and accuracy degradation? Without such a control, it is unclear whether the observed effects are attributable to the semantic content of personas (as the paper argues) or simply to the added context length and attentional interference. This is the paper's most significant gap, as it potentially undermines the causal interpretation of every main finding.

The paper reports flip rates and accuracy deltas, but these aggregate metrics make it difficult to assess the practical severity of the bias. Is the degradation concentrated in ambiguous, borderline items, or does it also affect scenarios where the correct emotional interpretation is unambiguous? A more granular analysis, for instance clustering flipped responses by scenario difficulty or emotional category, or using an LLM-as-a-judge to qualitatively characterize how responses shift (e.g., toward stereotypical emotional attributions vs. random errors), would substantially strengthen the contribution. The error taxonomy in Section 5.1 is a step in this direction but remains coarse; it classifies reasoning failures without examining the direction or systematicity of the emotional interpretation changes themselves.

The authors augment their existing 4-attribute intersectional personas with a fifth attribute (wealth, education, or disability) and, finding no significant additional effect, conclude that the original four dimensions are the appropriate ones. This reasoning is problematic. If the base persona already saturates the model's sensitivity to injected context, a plausible hypothesis given the degradation patterns in Section 5.2, then adding a fifth attribute would show no marginal effect regardless of its individual potency. A cleaner experiment would test each demographic attribute in isolation (single-attribute personas) and compare the resulting accuracy shifts, allowing the authors to identify which dimensions independently drive the observed biases.

The research question is timely and the experimental scope is commendable. However, the absence of a content-matched neutral control leaves the core causal claim, that persona-specific information drives differential emotional reasoning, inadequately supported. I would be more confident recommending acceptance if the authors addressed this confound directly.

---

### Official Review · Reviewer_Pfp6 · 2026-02-19
**Persistent profile information that affects emotional reasoning in LLMs**

**Rating:** 3
**Confidence:** 4

**Summary:**

This paper investigates how user memory (persistent profile information) affects emotional reasoning in LLMs. The authors evaluate 15 models on two validated emotional intelligence instruments, the Situational Test of Emotional Understanding (STEU) and a modified Situational Test of Emotion Management (STEM), under three conditions: no user profile, an advantaged user profile (high social capital), and a disadvantaged user profile (low social capital). They also construct 81 intersectional personas crossing gender, age, religion, and race to isolate demographic effects. Key findings are: (1) adding user memory systematically degrades emotional understanding performance, with disadvantaged profiles causing greater degradation and higher "flip rates" in several high-performing models; (2) demographic biases emerge across religion, gender, and age dimensions; (3) these biases persist when models give emotional advice (modified STEM). The paper includes ablation studies on memory injection methods (RAG vs. system prompt), additional demographic dimensions (wealth, education, disability), error analysis, and a preliminary DPO-based mitigation experiment on small models.

**Strengths:**

This paper sits squarely at the intersection of fairness, alignment, and personalization, the core concerns of the workshop. The personalization trap framing directly addresses how alignment procedures (memory-enhanced systems designed to be more helpful) can inadvertently embed social hierarchies into AI reasoning, which is precisely what the workshop aims to explore. The three-experiment progression is logically structured: Experiment 1 establishes that memory matters (RQ1), Experiment 2 isolates which demographic factors drive the effect (RQ2), and Experiment 3 shows the bias extends to actionable advice (RQ3). The use of validated psychometric instruments (STEU, STEM) rather than ad-hoc benchmarks adds credibility. The human annotation procedure for filtering persona-sensitive items (the two-phase quality gate) is particularly thoughtful; it ensures the ground truth answers genuinely shouldn't vary with persona, making any observed variation attributable to model bias. Testing 15 models spanning multiple architectures (Claude, Llama, DeepSeek, Qwen, Phi, Mistral, Command R) and sizes, including both standard and thinking variants, provides a comprehensive landscape view. The finding that reasoning-capable models show lower biases is an actionable insight. While limited to small models, the DPO fine-tuning experiment (Section 6) demonstrates that the identified biases are potentially addressable, moving the paper beyond pure diagnosis. The careful design (using persona-augmented but emotionally unrelated training data) avoids leaking test content.

**Weaknesses:**

It's well-established that irrelevant context in prompts degrades LLM performance, which is essentially a distraction or attention dilution effect. The paper's own ablation (Table 3) shows that even generic long context or random conversation history substantially degrades STEU performance (e.g., Claude 3.5 Haiku drops from 69% to 43–53%). This raises the question: how much of the observed degradation is specific to socially-biased personalization versus general context pollution? The paper needs a stronger control condition, such as injecting persona-length irrelevant text that contains no demographic or social information, to cleanly separate the distraction effect from the bias effect. Without this, the differential between advantaged and disadvantaged profiles (which is the actual bias signal) stands, but the overall drop may be less about personalization and more about prompt length.

Although the authors state that negative sentiment across personas is below 0.3, the disadvantaged profiles inherently describe hardship (e.g., "Sells plasma to afford interlibrary loan fees," "Rents a basement apartment with mold issues"). Even with low negative sentiment scores, these descriptions carry affective valence that could shift model behaviour through emotional priming rather than social bias per se. A disadvantaged persona described in neutral bureaucratic language ("annual income: $15,000, education: GED, occupation: part-time retail") might produce different results than the narrative-rich profiles used here. From my understanding, the paper doesn't control for this.

The three models fine-tuned (Qwen 0.5B, LLaMA 1B, Gemma 2B) are extremely small and already perform near random on STEU (Gemma 2B: 14% base → 14.29% after DPO on a 5-choice test). An improvement from 22% to 29.76% for LLaMA 1B is hard to interpret meaningfully when performance is still well below any useful threshold. The paper acknowledges this limitation but presents it as a positive result anyway. The mitigation section would be stronger as a brief future direction rather than a full section with a results table.

Finally, **there are typos** (e.g., "demsonstrates," "clasisfication," "labled," "ovlarger"), inconsistent formatting, and a duplicate figure caption ("Figure 4: Figure 4:"). The limitations section is unusually sparse and self-aware about this ("We could benefit from doing more error analysis"). For a venue like ICLR, the presentation needs polish.

---

### Official Review · Reviewer_TfLC · 2026-02-20
**Personalization Changes Emotional Reasoning and Introduces Measurable Demographic Disparities**

**Rating:** 4
**Confidence:** 4

**Summary:**

This paper studies how long-term user memory and personalization affect emotional reasoning in large language models. The authors evaluate 15 models using validated emotional intelligence benchmarks (STEU and a modified STEM task) and test whether adding user profiles to model memory changes emotional understanding and advice. They systematically vary demographic and socioeconomic attributes using controlled personas.

The results show that incorporating user memory significantly alters emotional reasoning performance. In many models, performance decreases when user profiles are introduced. Additionally, demographic disparities emerge: some models perform better for advantaged personas than disadvantaged ones, and these differences persist in emotional advice tasks. They also propose a mitigation strategy using Direct Preference Optimization (DPO). Fine-tuning on a curated dataset designed to reduce sensitivity to irrelevant persona information might improve emotional task performance and reduces demographic gaps.

**Strengths:**

- As LLMs increasingly use long-term memory, studying how personalization affects fairness in emotional reasoning is highly relevant.

- The paper evaluates number of models, includes intersectional personas, and studies both emotional understanding and emotional advice.

**Weaknesses:**

- The paper treats benchmark correctness as a proxy for fairness. Emotional fairness may require deeper analysis (e.g., tone, empowerment, agency framing), which is not explored.

- The connection to Bourdieu’s social capital is interesting but not fully operationalized in the experiments.

- The DPO experiment is promising but limited to smaller-scale models. It remains unclear whether similar improvements would hold for larger, state-of-the-art systems.

---

### Meta-Review · Area_Chair_a2mX · 2026-02-26

**Recommendation:** Reject
**Confidence:** 4

**Metareview:**

This paper studies how long-term user memory/personalization affects LLM emotional reasoning and advice, evaluating 15 models on validated EI benchmarks (STEU/STEM) under advantaged vs. disadvantaged profiles and intersectional personas, and exploring a DPO-based mitigation. Reviewers agree the topic is timely and well aligned with the workshop, and appreciate the breadth of experiments and the rigorous human annotation pipeline. However, they share a common concern about the core causal claim: adding long or irrelevant context substantially degrades performance, so the observed effects may reflect context length/attention dilution rather than persona semantics. I recommend the authors add a content-matched neutral control (persona-length text without demographic/affective content) and verify whether the demographic gaps persist under this control, which would substantially strengthen the paper.

---

### Decision · Program_Chairs · 2026-03-02

**Decision:**

Accept (Poster)

**Comment:**

The paper was originally submitted under the Main Track. While the paper is not ready to be accepted as a Main Track Paper, we find the work promising, and are giving an opportunity to the authors to instead get accepted in the Tiny/Short Track. Please submit a camera-ready version of up to 3 pages to comply with the Tiny/Short Paper requirements (more instructions on how to submit the camera-ready version will follow soon). The authors can also decide to withdraw if they prefer to not be accepted under the Tiny/Short Track.